# Prevalence of clinical signs of possible serious bacterial infection and mortality associated with them from population-based surveillance of young infants from birth to 2 months of age

**Dhruv Puri[1], Yasir Bin Nisar🔗[1]\*, Antoinette Tshefu[2], Adrien Lokangaka Longombe[2], Fabian Esamai[3], Irene Marete[3], Adejumoke Idowu Ayede[4], Ebunoluwa A. Adejuyigbe[5], Robinson D. Wammanda[6], Shamim Ahmad Qazi[7], Rajiv Bahl[1]**

1 Department of Maternal, Neonatal, Child and Adolescent Health and Ageing, World Health Organization, Geneva, Switzerland, 2 Department of Community Health, Kinshasa School of Public Health, Kinshasa, DR Congo, 3 Department of Child Health and Paediatrics, School of Medicine, Moi University, Eldoret, Kenya, 4 College of Medicine, University of Ibadan, and University College Hospital, Ibadan, Nigeria, 5 Department of Paediatrics and Child Health, Obafemi Awolowo University, Ile-Ife, Nigeria, 6 Department of Community Medicine, Ahmadu Bello University Teaching Hospital, Ahmadu Bello University, Zaria, Nigeria, 7 Child and Newborn Health Consultant, Geneva, Switzerland

\* nisary@who.int

## Abstract

### Background

Community-based data on the prevalence of clinical signs of possible serious bacterial infection (PSBI) and the mortality associated with them are scarce. The aim was to examine the prevalence for each sign of infection and mortality associated with infants in the first two months of life, using community surveillance through community health workers (CHW).

### Methods

We used population-based surveillance data of infants up to two months of age from the African Neonatal Sepsis Trial (AFRINEST). In this study, CHWs visited infants up to 10 times during the first two months of life at five sites in three sub-Saharan African countries. CHW assessed the infant for signs of infection (local or systemic) and referred infants who presented with any sign of infection to a health facility. We used a longitudinal analysis to calculate the risk of death associated with the presence of a sign of infection at the time of the visit until the subsequent visit.

### Results

During the first two months of their life, CHWs visited 84,759 live-born infants at least twice. In 11,089 infants (13.1%), one or more signs of infection were identified, of which 237 (2.1%) died. A sign of infection was detected at 2.1% of total visits. In 52% of visits, infants had one or more sign of systemic infection, while 25% had fast breathing in 7–59 days

**Data Availability Statement:** The complete de-identified patient data set cannot be shared publicly due to institutional restrictions. However, upon receipt of a proposal with rationale for use and agreement to abide by WHO data access policies, data can be obtained from the primary author YBN (nisary@who.int) or Nicole Grillon (grillonn@who.int), Department of Maternal, Newborn, Child and Adolescent Health and Ageing, World Health Organization, Geneva.

**Funding:** The study was funded by a grant from the Bill & Melinda Gates Foundation to WHO, which sponsored the study.

**Competing interests:** YBN and RB are WHO employees. All other authors declare no competing interests.

period and 23% had a local infection. All signs of infection, including multiple signs, were more frequently seen in the first week of life. The risk of mortality was very low (0.2%) for local infections and fast breathing in 7–59 days old, it was low for fast breathing 0–6 days old (0.6%), high body temperature (0.7%) and severe chest indrawing (1.0%), moderate for low body temperature (4.9%) and stopped feeding well/not able to feed at all (5.0%) and high for movement only when stimulated or no movement at all (10%) and multiple signs of systemic infection (15.5%). The risk of death associated with most clinical signs was higher (1.5 to 9 times) in the first week of life than at later age, except for low body temperature (4 times lower) as well as high body temperature (2 times lower).

## Conclusion

Signs of infections are common in the first two months of life. The mortality risk differs with clinical signs and can be grouped as very low (local infections, fast breathing 7–59 days), low (fever, severe chest indrawing and fast breathing 0–6 days), moderate (low body temperature and stopped feeding well/not able to feed at all) and high (for movements only on stimulation or no movements at all and multiple signs of infection). New treatment strategies that consider differential mortality risk could be developed and evaluated based on these findings.

## Clinical trial registration

The trial was registered with Australian New Zealand Clinical Trials Registry under ID ACTRN 12610000286044.

## Background

In 2018, an estimated 5.3 million under-five-year-old deaths occurred globally and 2.5 million died in the first four weeks of life [1]. One-third of these neonatal deaths occurred on the first day of life and three-quarters in the first week. Most deaths occurred in low-and middle-income countries (LMIC) with sub-Saharan Africa leading the world in both neonatal and under-five mortality rates [1]. Prematurity, birth asphyxia and neonatal infections are the major causes of neonatal deaths globally [1,2]. Maternal sepsis during pregnancy, labour and post-partum and post caesarean section infections also result in neonatal deaths in low resource settings [3–5]. The overall burden of possible serious bacterial infection (PSBI) in young infants of LMICs is substantial [6,7]. A systematic review and meta-analysis that included data from 22 studies, for 259,944 neonates and 20,196 PSBI cases reported pooled estimate of PSBI incidence risk as 7.6% and a case fatality rate (CFR) of 9.8% (3). It estimated that 6.9 million cases of PSBI needed treatment in 2012. In a large multicentre prospective study in Argentina, Guatemala, India, Kenya, Pakistan, Zambia, the incidence of PSBI ranged from 3% in Zambia to 36% in Pakistan in a cohort of 248,539 live births in the first six weeks of life. The incidence of PSBI during the first 6 weeks of life varied 10 fold between sites [7].

Over the past two decades, there has been substantial simplification in identifying PSBI using symptoms and clinical signs. In the 1990s, a multicentre study conducted in Ethiopia, The Gambia, The Philippines, Papua New Guinea identified 13 predictors of serious infection, which should be referred to a hospital [8]. These signs (convulsions, fast breathing defined as the respiratory rate of 60 breaths per minute or more, severe chest indrawing, nasal flaring,

grunting, bulging fontanelle, pus drainage from the ear, temperature $\geq 37.5˚C$ or $<35.5˚C$ or feels cold, umbilical redness extending to the skin, many or severe skin pustules, lethargy or unconsciousness, less than normal movement) were included in the first Integrated Management of Childhood Illness (IMCI) chart booklet in 1997 to identify PSBI in 7–59 days old young infants [9]. It was challenging to train and maintain first-line health workers' skills for all signs. In 2008, another study conducted in Bangladesh, Bolivia, Ghana, India, Pakistan, and South Africa found that seven signs had a sensitivity of 85% and specificity of 75% to identify PSBI in the first week of life and sensitivity of 74% and specificity of 79%) in 7-59-day-old infants [10]. The revised IMCI chart booklet in 2014 used seven signs (not feeding well, convulsions, movement only when stimulated or no movement at all, fast breathing, severe chest indrawing, the temperature of $\geq 37.5˚C$ or $<35.5˚C$) to identify PSBI in young infants for a referral to a hospital [11].

Unfortunately, many families in low resource settings do not accept referral advice for a hospital [12–15]. Four trials were conducted in Asia and Africa to evaluate the effectiveness of simplified antibiotic regimens on an outpatient basis when the referral was not feasible [16–19]. These data showed that young infants with only fast breathing had a negligible death rate and those categorized as clinical severe infection (stopped feeding well, movement only when stimulated, severe chest indrawing, axillary temperature of $\geq 38˚C$ or $<35.5˚C$) had a death rate of 1–2%. Critically ill young infants (convulsions, not able to feed at all, no movement at all) were not included in these studies. This data contributed to a World Health Organization guideline to manage PSBI when a referral was not feasible [20]. It recommended fast breathing in 7–59 days old young infants to be treated with oral amoxicillin on an outpatient basis without a referral, and clinical severe infection to be treated with injectable gentamicin plus oral amoxicillin on an outpatient basis when the referral was not feasible [20].

Community-based data using specific clinical signs for identifying infections and associated mortality in newborns and young infants are scarce. Bang et al from India reported that identification by community health workers (CHW) of any two out of seven signs simultaneously (reduced or stopped sucking; weak or no cry; limbs becoming limp; vomiting or abdominal distension; baby cold to touch; severe chest indrawing; umbilical infection) predicted a sepsis death in a newborn [21]. Another study from Nepal reported CHW experience of identifying PSBI from the community and identified two signs concurrently having a higher CFR [22]. Several studies, validated the clinical signs used by CHW to identify serious infections in the community from Bangladesh [23–25] and one from Nepal validated signs for umbilical sepsis only at the community level [26]. None of the studies reported mortality for each sign separately and all reported a much smaller number of newborns followed in the community, compared to AFRIcan NEonatal Sepsis Trial (AFRINEST) [27,28]. Secondly, all studies were conducted in Asia.

In the AFRINEST community surveillance was carried out by CHW who visited each newborn at home several times up to two months of age to identify sick young infants with signs of PSBI We used this population-based data to conduct secondary analysis to assess the prevalence of each clinical sign and its association with mortality.

## Methods

### Study design

We used population-based surveillance data from the AFRINEST which was a prospective randomized controlled trial over five sites in three sub-Saharan African countries–Democratic Republic of the Congo (DRC), Kenya, and Nigeria [16,17]. The methodology of the trial has been published [27,28]. Briefly, two trials were conducted—one trial evaluated the simplified

antibiotic regimens for management of young infants up to 2 months of age with clinical severe infection when hospital admission was not feasible [17], and the other compared oral antibiotic and injectable plus oral antibiotic therapies for fast breathing in young infants [16].

## Data collection and follow-up

In AFRINEST, we established a population-based surveillance system at all five study sites to identify pregnant women, collect information on the birth status and then followed live births for up to two months of age through community health workers (CHW). CHWs in DRC and Kenya and Community Health Extension Workers (CHEWs) in Nigeria conducted follow up visits scheduled on days 1, 3, 7, 14, 21, 28, 35, 42, 49, and 59 after birth and assessed young infants for signs of infection (Panel 1). The CHEWs had 2–3 years formal training compared to CHWs who had been trained for a few weeks [28]. Infants with any sign of infection were referred to a health facility or hospital for further assessment and management. Study nurses at health facilities examined sick infants identified by CHW/CHEWs to validate signs of infection. Infants with any sign of infection were referred to hospital for treatment and families who refused referral were offered simplified antibiotic regimens on an outpatient basis [27,28]. Quality assurance was ensured by close monitoring and periodic training of all CHWs/ CHEWs in WHO/UNICEF caring for newborns in the community training module and the study procedures [27–29].

## Study exposure variables and outcome

At each visit, CHWs/CHEWs assessed young infants for signs of infection (Panel 1) and recorded their observations in a case report form. These signs, which we considered as study exposure variables, were classified into three mutually exclusive categories of 'no sign of infection', 'signs of local infection', and 'fast breathing in 7–59 days old', 'signs of systemic infection' (Panel 1).

The study outcome was the status of the child as alive or death (all-cause) recorded by CHWs/CHEWs during each visit. The association of other information such as study site, sex of the infant, maternal age, place of delivery and weight at birth with the study outcome was investigated.

## Ethical considerations

The AFRINEST study was approved by the site institutional ethics review committees and the WHO Research Ethics Review Committee. We obtained written and witnessed informed consent from parents/caregiver.

**Panel 1. Classification for this analysis using signs of infection in young infants aged 0 to 59 days as recorded by community health workers and community health extension workers.**

- Local infection as defined by pus in the umbilicus, skin, or eye
- Fast breathing as defined by respiratory rate of 60 or more breaths per minute (in the first 7–59 days of life)

*Signs of Systemic Infection*

- Fast breathing as defined by respiratory rate of 60 or more breaths per minute (in the first 0–6 days of life)
- Stopped feeding well or unable to feed according to the mother
- Reported convulsions
- High body temperature defined as an axillary temperature of 37.5˚C and above
- Low body temperature defined as an axillary temperature of below 35.5˚C
- Severe chest indrawing
- Movement only on stimulation or no movement at all

### Statistical analysis and definitions

Data analysis was performed using Stata™, Version 14 (Stata-Corp, College Station, TX, USA). For analysis purposes, if a sign was recorded as missing at a particular visit in the database, it was inferred to be the same as the immediately preceding visit. We defined the period between two visits as an infant-period. This was usually seven days (between two consecutive visits by a CHW/CHEW to assess the same young infant), except in the case of the first week of life (shorter than 7 days) and the last 10 days of the second month (longer than 7 days).

We selected infants up to 2 months of age with at least two visits by CHWs/CHEWs. Missing outcomes were inputted under the assumption that infants whose outcome was known in a later visit could be assumed to be alive at prior visits. Young infants whose information on signs of infection at either of the two visits immediately preceding outcome was not available were excluded from the analysis. We used a longitudinal analysis to calculate the risk of death associated with the presence of a sign of infection at the time of visit until the subsequent visit expressed as a percentage. Presence of each sign of infection was calculated as the number of young infants with a specific sign of infection divided by the total number of infant-periods followed and expressed as per 1000 infant-periods.

## Results

We registered a total of 95299 pregnant women from all sites. Of these, we identified 85592 (89.8%) live births and among live births, 84759 infants (99%) were visited at least two times (of a maximum possible of 10) by CHWs/CHEWs and were included in analyses. One or more signs of infection were identified at least once in 11,089 infants (13.1%), of which 237 (2.1%) died. Of the 73670 infants with no sign of infection, 350 (0.5%) died (Fig 1).

Fig 2 shows (a) presence of any sign of infection, and (b) the proportion of single or multiple signs of infection among young infants at the time of visits. A sign of infection was detected at 2.1% of visits. In 52% of visits, infants had one or more sign of systemic infection, while 25% had fast breathing in 7–59 days period and 23% had a local infection.

All signs of infection including local infections, single and multiple signs of systemic infection were frequently seen in the first week of life (Table 1). Local infections were most common in the second (16/1000) and third week (14/1000) of life, fast breathing (10-17/1000) and stopped feeding well/not able to feed at all (1/1000) were commonest in the first three weeks of life and severe chest indrawing was more frequent in the first 4 weeks of life (2-4/1000). High body temperature (9-10/1000), low body temperature (2-8/1000) and multiple signs (6/1000) were observed most commonly in the first week of life. Movement only on stimulation/no movement at all and convulsions was infrequently seen.

Fig 3 shows a clear gradation of risk of mortality from negligible with no signs followed by local infections, fast breathing, high body temperature, severe chest indrawing, whereas a higher risk of death was observed for low body temperature, and stopped feeding well/not able to feed at all. The highest risk of death was observed with the combination of signs of systemic infection. Additionally, we also observed a time trend for risk of death. For local infections, fast breathing, severe chest indrawing, stopped feeding well/not able to feed at all, the risk of death was highest in the first week of life. For multiple signs, the risk of death was highest in the first two weeks of life.

Table 2 shows that the signs have different levels of mortality by age. The risk of mortality was very low (0.2%) for local infections and fast breathing in 7–59 days old, it was low for fast breathing 0–6 days old (0.6%), high body temperature (0.7%) and severe chest indrawing (1.0%), moderate for low body temperature (4.9%) and stopped feeding well/not able to feed at all (5.0%) and high for movement only when stimulated or no movement at all (10%) and multiple signs of systemic infection (15.5%). The risk of death associated with most clinical signs

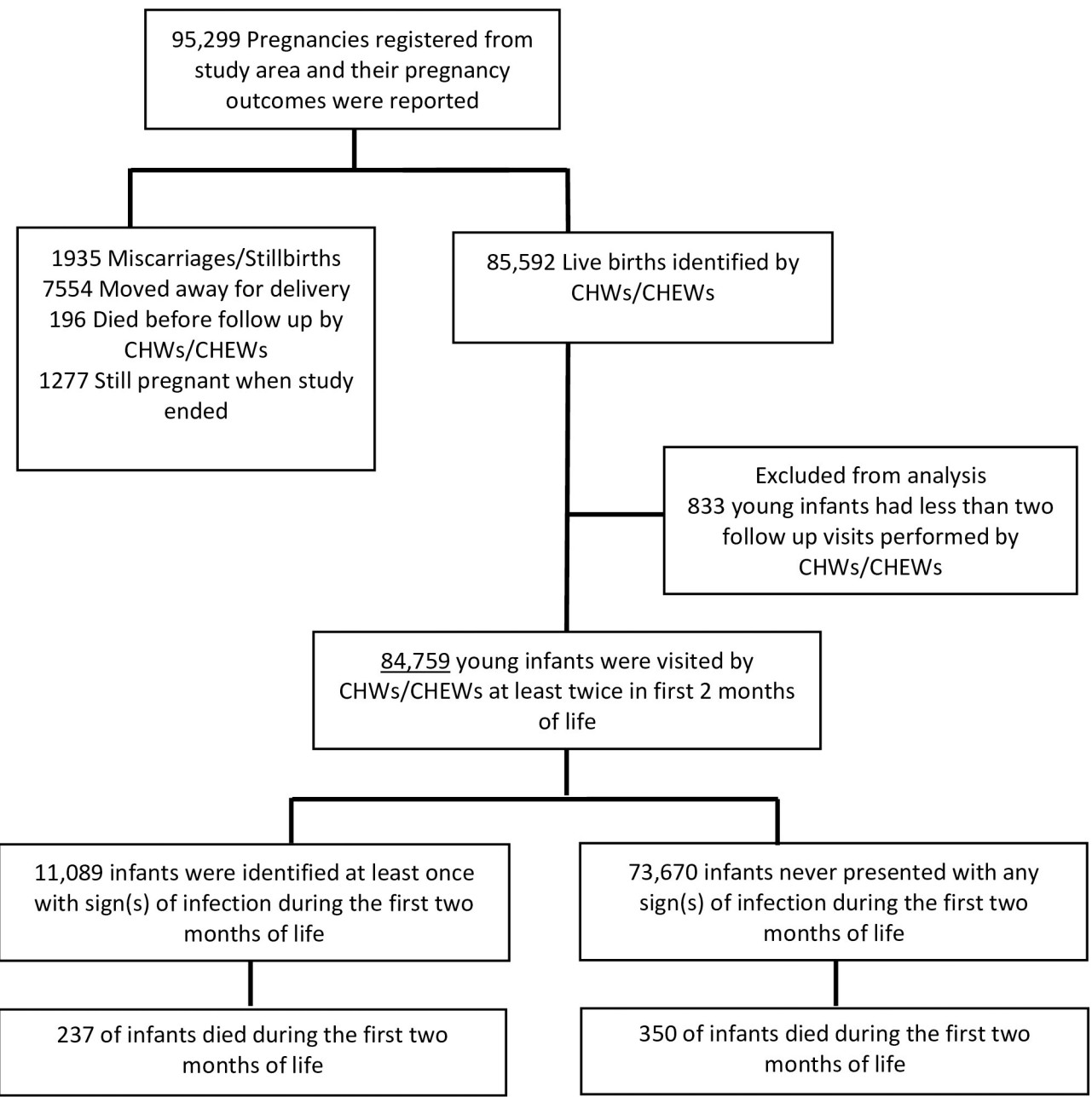

**Fig 1. Flow diagram of the exclusion criteria to establish the study population.** CHW: Community health worker. CHEW: Community health extension worker.

was higher (1.5 to 9 times) in the first week of life than at later age, except for low body temperature (4 times lower) as well as high body temperature (2 times lower).

## Discussion

### Key findings

The population-based surveillance data of young infants from three sub-Saharan African countries has several important findings. First, one in seven young infants had at least one

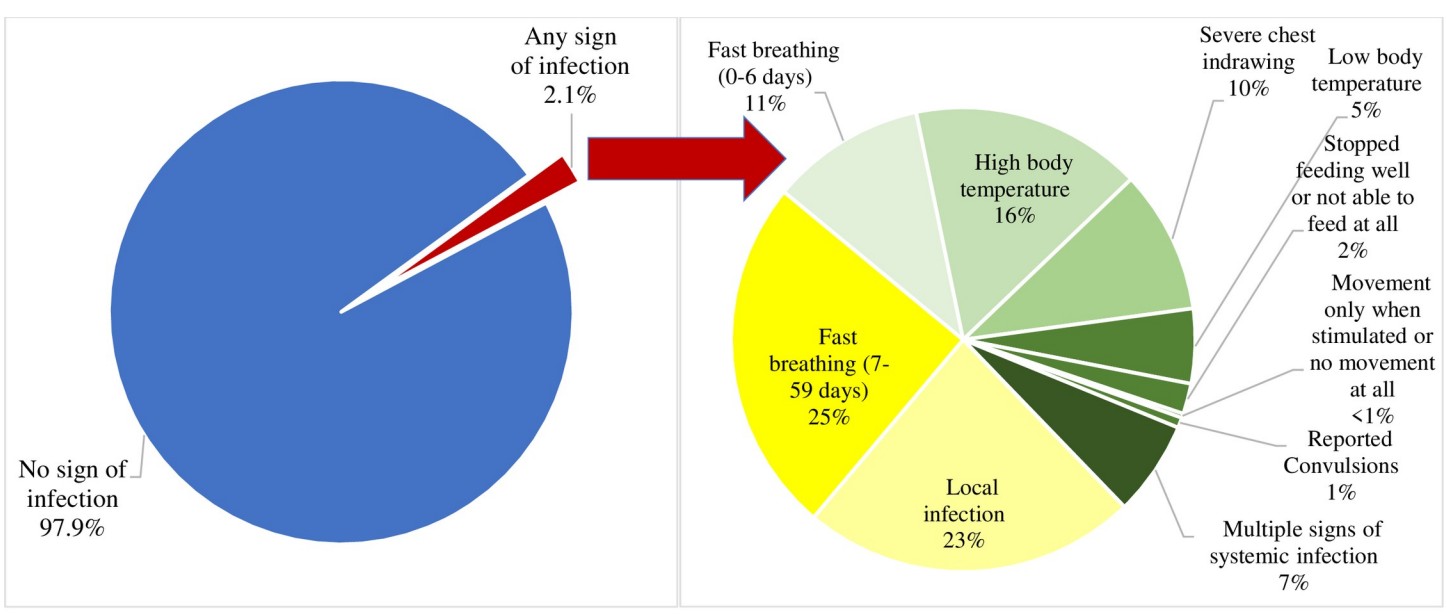

Fig 2a                                                    Fig 2b

**Fig 2.** a) Cumulative presence of any sign of infection in the first 2 months of life during follow-up visits by infant-periods, b) Proportion of individual and multiple signs of infection in infants who had any sign of infection. There were 616,421 infant-periods of follow up and in 13,260 (2.1%) of infant-periods, the infant had a sign of infection.

**Table 1. Frequency of signs of infection by age of assessment (per 1000 infants) reported by CHWs/CHEWs\*.**

| Number of infants | Young infants assessed after birth on | | | | | | | | | |
|---|---|---|---|---|---|---|---|---|---|---|
| | Day 1 47471 | Day 3 56279 | Day 7 63817 | Day 14 69057 | Day 21 71905 | Day 28 74024 | Day 35 76202 | Day 42 78189 | Day 49 79407 | Day 59 79008 |
| | n (per 1000 infants) | n (per 1000 infants) | n (per 1000) | n (per 1000) | n (per 1000) | n (per 1000) | n (per 1000) | n (per 1000) | n (per 1000) | n (per 1000) |
| Local infections† | 162 (4) | 392 (7) | 876 (14) | 822 (12) | 442 (6) | 236 (3) | 154 (2) | 97 (1) | 59 (1) | 21 (<1) |
| Fast breathing (7–59 Days) | NA‡ | NA | 1001 (16) | 933 (14) | 603 (8) | 347 (5) | 215 (3) | 152 (2) | 123 (2) | 59 (1) |
| **Signs of systemic infection** | | | | | | | | | | |
| Fast breathing (0–6 Days) | 472 (10) | 968 (17) | NA | NA | NA | NA | NA | NA | NA | NA |
| High body temperature | 418 (9) | 537 (10) | 336 (5) | 275 (4) | 182 (3) | 124 (2) | 106 (1) | 99 (1) | 9 (<1) | 59 (1) |
| Severe chest indrawing | 107 (2) | 226 (4) | 204 (3) | 214 (3) | 192 (3) | 134 (2) | 115 (2) | 75 (1) | 73 (1) | 22 (<1) |
| Low body temperature | 391 (8) | 133 (2) | 85 (1) | 58 (1) | 25 (<1) | 14 (<1) | 11 (<1) | 10 (<1) | 7 (<1) | 3 (<1) |
| Stopped feeding well or not feeding at all | 58 (1) | 79 (1) | 37 (1) | 47 (1) | 30 (<1) | 17 (<1) | 13 (<1) | 9 (<1) | 11 (<1) | 8 (<1) |
| Movement only when stimulated or no movement at all | 8 (<1) | 11 (<1) | 11 (<1) | 6 (<1) | 3 (<1) | 4 (<1) | 3 (<1) | 2 (<1) | 2 (<1) | 3 (<1) |
| Reported convulsions | 20 (<1) | 17 (<1) | 12 (<1) | 12 (<1) | 9 (<1) | 14 (<1) | 5 (<1) | 6 (<1) | 0 (0) | 0 (0) |
| Multiple signs of systemic infection | 278 (6) | 302 (6) | 118 (2) | 97 (1) | 72 (1) | 54 (1) | 27 (0) | 36 (0) | 23 (0) | 14 (0) |

\* CHW: Community health worker; CHEW: Community health extension worker.

† 3261 local infections were reported, 498 (15.3%) umbilical infections, 1808 (55.4%) skin infections, 817 (25.1%) eye infections and 138 (4.2%) mixed local infections.

‡NA: Not applicable.

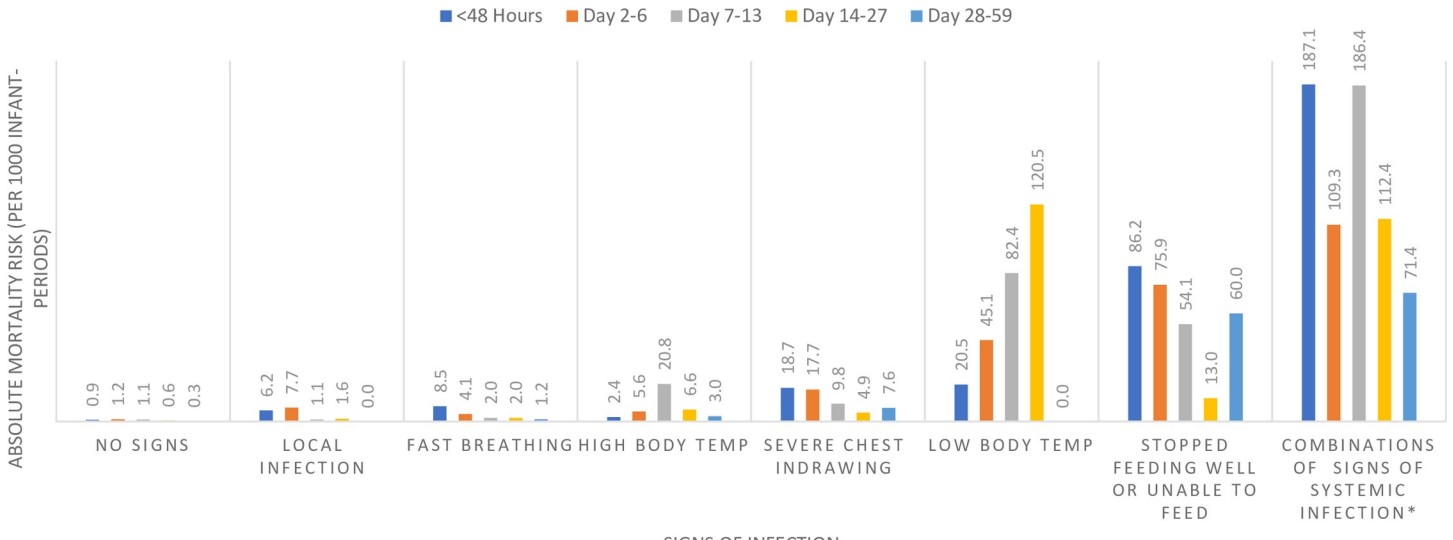

**Fig 3. Distribution of risk of mortality associated with signs of infection† (per 1000 infant-periods) in the first two months of life.** † 'Reported convulsions' was excluded from the figure as no deaths occurred in young infants with only convulsions. 'Movement only when stimulated or no movement at all' was excluded due to low prevalence (<20 cases) per visit. * These children had multiple signs of systemic infection.

episode of either local or systemic infection when assessed by a CHW/CHEW at any visit during the first two months of life. Second, nearly half of the sick young infants identified at the community level had either fast breathing or local infections, followed by fever and chest indrawing that were seen more often than other signs such as stopped feeding well/not able to feed at all, low body temperature and multiple signs. CHWs didn't find many young infants with movement only on stimulation/no movement at all or convulsions. Third, All signs of infection were frequently seen in the first week of life, but fever, low body temperature and multiple signs were most common in this age group. Fast breathing, chest indrawing,

**Table 2. Number of deaths by infant-periods of follow up and risk of death (%) by clinical signs stratified by 0–6 days and 7–59 days of life.**

| | 0–6 days of life | | 7–59 days of life | | Total | |
|---|---|---|---|---|---|---|
| Clinical assessment by CHWs/CHEWs | Deaths/infants-period | Risk of death % (95% CI) | Deaths/infants-period | Risk of death % (95% CI) | Deaths/infants-period | Risk of death % (95% CI) |
| No Sign of infection | 245/53279 | 0.5 (0.4, 0.5) | 105/549882 | 0.02 (<0.1, <0.1) | 350/603161 | 0.1 (0.1, 0.1) |
| Local infections | 4/453 | 0.9 (0.2, 2.2) | 3/2645 | 0.1 (0.1, 0.3) | 7/3098 | 0.2 (0.1, 0.5) |
| Fast breathing (7–59 days) | NA | NA | 6/3293 | 0.2 (0.1, 0.4) | 6/3293 | 0.2 (0.1, 0.4) |
| **Signs of Systemic Infection** | | | | | | |
| Fast breathing (0–6 days) | 8/1429 | 0.6 (0.2, 1.1) | NA | NA | 8/1429 | 0.6 (0.2, 1.1) |
| High body temperature | 4/952 | 0.4 (0.1, 1.1) | 11/1184 | 0.9 (0.5, 1.7) | 15/2136 | 0.7 (0.4, 1.2) |
| Severe chest indrawing | 6/331 | 1.8 (0.7, 3.4) | 7/993 | 0.7 (0.3, 1.4) | 13/1324 | 1.0 (0.5, 1.7) |
| Low body temperature | 14/504 | 2.8 (1.5, 4.6) | 20/185 | 10.8 (6.7, 16.2) | 34/689 | 4.9 (3.4, 6.8) |
| Stopped feeding well or not able to feed at all | 11/130 | 8.5 (4.3, 14.6) | 3/151 | 2.0 (0.4, 5.7) | 14/281 | 5.0 (2.7, 8.2) |
| Movement only when stimulated or no movement at all | 2/14 | 14.3 (1.8, 42.8) | 2/26 | 7.7 (0.9, 25.1) | 4/40 | 10.0 (2.8, 23.7) |
| Reported convulsions | 0/37 | - | 0/58 | - | 0/95 | - |
| Multiple signs of systemic infection | 85/496 | 17.1 (13.9, 20.7) | 51/379 | 13.5 (10.2, 17.3) | 136/875 | 15.5 (13.2, 18.1) |

NA: Not applicable.

movement only on stimulation/no movement at all and feeding difficulty were common in the first three weeks of life. Fourth, there was a clear gradation of risk of death with various signs. It was a very low risk of death for local infections and fast breathing (7–59 days), low risk for fast breathing (0–6 days), high body temperature and severe chest indrawing, moderate risk for low body temperature and stopped feeding well/not able to feed at all, and high risk for movement only when stimulated or no movement at all and multiple signs of infection. Finally, the risk of death associated with most clinical signs was higher in the first week of life than at later age, except for low body temperature as well as high body temperature.

Our data showed a higher prevalence of signs of infection during the first two months of age compared to an earlier systematic review and meta-analysis of 22 studies, which reported incidence risk of PSBI as 7.6%. The incidence risk was slightly higher (8.4%) in studies with active follow-up using seven signs of infection (excluding local infection signs) [6]. Our higher prevalence could be due to the inclusion of local infection signs, which were included because it is a referral sign at the community level [30]. Second, we conducted active surveillance through home visits by CHWs/CHEWs up to 10 times during the first two months of life. In contrast, another prospective multi-country study reported the incidence of PSBI range from 3% to 36% in different countries in the first six weeks of life [7].

## Implications of results

First, we found a higher frequency of signs of systemic infection during the first week of life. Also, we observed the highest risk of death in the same period and higher rates of death for most systemic infection signs in the same period. This fits in well with the global data reporting one-third of neonatal deaths occurring on the first day of life and three-quarters in the first week of life [1]. This highlights the need for identifying these sick newborns as soon as possible to provide them with appropriate treatment. The WHO postnatal guidelines recommend two home visits by a CHW in the first week of life to counsel mothers on the care of the newborn and identify sick newborns [31]. Although the reported median coverage of postnatal home visits for newborns has increased from 30% in 2009–2013 to 68% in 2014–2019, it is still not enough [32]. Identifying sick young infants is not enough, getting them to well-trained health workers and health facility to get appropriate treatment is critical. Unfortunately, data shows that a large number of families do not accept referral advice to go to a hospital [16–19,33,34]. Families refuse referral due to cost of travel and hospitalization, distance, religious and cultural beliefs, lack of permission from husband or family elders, concerns around quality of care at the hospital, poor attitudes of health workers, and lack of child care/other logistical barriers [35–37]. Thus more efforts are needed to overcome these barriers through family counselling, community engagement, improving the quality of care at the health facilities and reduce the number of referrals by providing care nearer to people's homes.

Second, a young infant with low body temperature and high body temperature should receive care throughout the first two months of life. Although the frequency of low body temperature and high body temperature was higher in the first week of life, one cannot rule out the environmental factors as its reason. Newborns are more likely to be exposed to low environmental temperatures in the first week of life and conversely, some may be wrapped much more so in the first week of life. But having low or high body temperature after the first week of life is more likely to be linked to an infection. The risk of death was four times higher for low body temperature after the first week of life and two times higher with high body temperature after the first week of life, so it needs more attention. A systematic review based on 10 community studies in Southeast Asia found that the prevalence of neonatal hypothermia ranged from 11% to 92%, whereas in hospital-based studies from Africa it ranged from 8% to 85%

[38]. Poor management of neonatal hypothermia may lead to severe sepsis and death [39,40]. CFR for hypothermia ranges from 8.5% to 52% [41–46].

Third, our analyses showed mortality risk differ by each clinical sign of infection. In the 1997 and 2014 IMCI chart booklets, each sign included in the algorithm was given equal weight to identify sick young infants for referral [9,11]. The 2015 WHO PSBI guideline removed fast breathing (7–59 days) from the referral advice, but the other seven signs were all given equal weight for referral recommendation [20,47]. We could find one study in infants up to 6 weeks of life with PSBI signs that collected data from health facilities, which reported CFR for individual signs, 1% with high fever, 18% for hypothermia, 22% with feeding problem/stopped sucking or feeding, 25% with convulsions and 27% with multiple signs [7]. This data allows us to hypothesize that there is further room for improvement in risk stratification for management of PSBI in the current WHO guideline, so that the low mortality risk signs (fever, any fast breathing including those in 0–6 days of age and severe chest indrawing) may not need referral to a hospital. Additionally, those with multiple signs of systemic infection need more attention, irrespective of PSBI sub-categorization. More research is needed to address these issues, which can inform decisions about further improving PSBI management.

Fourth, high early neonatal mortality rate is also linked with complications in pregnancy, labour and post-partum, so there it is essential to timely identify high-risk pregnancies and manage them in suitable high dependency and critical care settings in low and middle-income countries [48,49], which is challenging. Finally, prompt identification and diagnosis of neonatal sepsis require antibiotics and supportive treatment. Inappropriate antibiotic use contributes to antimicrobial resistance, which is quite widespread even in the high-income countries and is one of the greatest public health challenges of the 21st century [50]. Antibiotic stewardship is essential and health professionals treating sepsis should have the knowledge and follow the recommended antibiotic practices in light of the increasing antimicrobial resistance and avoid using unsuitable treatments [51,52].

### Strengths

Our study had several strengths. First, we used population-based surveillance data of young infants collected at five rural sites from three sub-Saharan African countries. Second, it was conducted in a large population from which community-level data was collected prospectively to calculate the risk of death and mortality rates for individual signs of infection in infants up to two months of age in sub-Saharan Africa. Third, CHWs/CHEWs were able to complete at least two postnatal visits in 99% of live births. Fourth, we used longitudinal analyses to evaluate the effect of a sign of infection on survival status during the subsequent visit and used infant-periods rather than individual infants as the unit of comparison allowing for risk of death to be stratified by a visit.

### Limitations

However, our study also had some limitations. First, no laboratory investigations or radiological studies were performed to confirm the diagnosis. Second, we did not validate the clinical findings of CHWs/CHEWs. Third, CHWs/CHEWs did not collect information on treatment status. Lastly, we did not collect information on environmental and genetic factors, congenital malformations and other comorbidities, or breastfeeding patterns with which to adjust our regression model.

### Conclusion

Signs of infections are common in the first two months of life. The mortality risk differs with clinical signs and can be grouped as very low (local infections, fast breathing 7–59 days), low

(fever, severe chest indrawing and fast breathing 0–6 days), moderate (low body temperature and stopped feeding well/not able to feed at all) and high (for movements only on stimulation or no movements at all and multiple signs of infection). New treatment strategies that consider differential mortality risk could be developed and evaluated based on these findings.

## Author Contributions

**Conceptualization:** Dhruv Puri, Yasir Bin Nisar, Antoinette Tshefu, Adrien Lokangaka Longombe, Fabian Esamai, Irene Marete, Adejumoke Idowu Ayede, Ebunoluwa A. Adejuyigbe, Robinson D. Wammanda, Shamim Ahmad Qazi, Rajiv Bahl.

**Data curation:** Dhruv Puri.

**Formal analysis:** Dhruv Puri, Yasir Bin Nisar.

**Investigation:** Yasir Bin Nisar, Antoinette Tshefu, Adrien Lokangaka Longombe, Fabian Esamai, Irene Marete, Adejumoke Idowu Ayede, Ebunoluwa A. Adejuyigbe, Robinson D. Wammanda, Shamim Ahmad Qazi, Rajiv Bahl.

**Methodology:** Yasir Bin Nisar, Antoinette Tshefu, Adrien Lokangaka Longombe, Fabian Esamai, Irene Marete, Adejumoke Idowu Ayede, Ebunoluwa A. Adejuyigbe, Robinson D. Wammanda, Shamim Ahmad Qazi, Rajiv Bahl.

**Writing – original draft:** Dhruv Puri, Yasir Bin Nisar, Shamim Ahmad Qazi, Rajiv Bahl.

**Writing – review & editing:** Dhruv Puri, Yasir Bin Nisar, Antoinette Tshefu, Adrien Lokangaka Longombe, Fabian Esamai, Irene Marete, Adejumoke Idowu Ayede, Ebunoluwa A. Adejuyigbe, Robinson D. Wammanda, Shamim Ahmad Qazi, Rajiv Bahl.

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
