## [Decision Letter · Decision Letter 0]

8 Jan 2021

PONE-D-20-16849

Prevalence of clinical signs of possible serious bacterial infection and mortality associated with them from a population-based surveillance of young infants from birth to 2 months of age

PLOS ONE

Dear Dr. NIsar,

Thank you for submitting your manuscript to PLOS ONE. After careful consideration, we feel that it has merit but does not fully meet PLOS ONE’s publication criteria as it currently stands. Therefore, we invite you to submit a revised version of the manuscript that addresses the points raised during the review process.

We look forward to receiving your revised manuscript.

Kind regards,

Claudia Marotta

Academic Editor

PLOS ONE

Additional Editor Comments:

dear authors follow reviewer suggestion to improve your paper

Reviewers' comments:

Reviewer's Responses to Questions

**Comments to the Author**

1. Is the manuscript technically sound, and do the data support the conclusions?

Reviewer #1: Yes

Reviewer #2: Yes

2. Has the statistical analysis been performed appropriately and rigorously? 

Reviewer #1: Yes

Reviewer #2: Yes

3. Have the authors made all data underlying the findings in their manuscript fully available?

Reviewer #1: Yes

Reviewer #2: Yes

4. Is the manuscript presented in an intelligible fashion and written in standard English?

Reviewer #1: Yes

Reviewer #2: Yes

5. Review Comments to the Author

Reviewer #1: I read the manuscript in 1 hour. I'm enthusiastic about the manuscript. I find it well wrote and with important topic.

Below my suggestions

1. Introduction: well done. underline better also the role of maternal cesarian section infection and how it is important also for maternal and neonatal outome expecially in low setting (see and cite this paper from Sierra Leone Di Gennaro F, Maternal caesarean section infection (MACSI) in Sierra Leone: a case-control study. Epidemiol Infect. 2020 Feb 27;148:e40. doi: 10.1017/S0950268820000370. PMID: 32102721; PMCID: PMC7058652.) and how maternal and neonatal outcomes are strongly related.

2. Methods well wrote

3. Results: no comment

4. Discussion: underline the lack of microbiological test for the identification of responsible germs and a better antibiotic therapy and identification of resistance. I believe there is a need to advocate on the development of microbiology in these contexts. Underline also the need to develop an Intensive Care unit for maternal and neonatal ( see and cite this experience

Epidemiology, Outcomes, and Risk Factors for Mortality in Critically Ill Women Admitted to an Obstetric High-Dependency Unit in Sierra Leone. Am J Trop Med Hyg. 2020 Nov;103(5):2142-2148. doi: 10.4269/ajtmh.20-0623. PMID: 32840199; PMCID: PMC7646769. and Cost-Utility of Intermediate Obstetric Critical Care in a Resource-Limited Setting: A Value-Based Analysis. Ann Glob Health. 2020 Jul 20;86(1):82. doi: 10.5334/aogh.2907. PMID: 32742940; PMCID: PMC7380057.)

Also highlithed the need for doctors to train themselves on the topics of antibiotic therapy, antimicrobial stewardship because the role of sepsis in neonatal and maternal deaths is very high (doi: 10.1016/j.jgar.2020.08.022. Epub ahead of print. PMID: 32971291.)

Reviewer #2: Few cocnerns:

1- Hypothermia is defined in your study as < 35.5, and

but the WHO its publication as well as in US NWN nurseries and defined by AAP, hypothermia in neonates is < 36.5 (few centers refer to 36.3). Please expailn why you use lower temp Also, if you use WHO definition as listed below, tjhat can completely change incidence of morbidity among the population you visited:

https://apps.who.int/iris/bitstream/handle/10665/60042/WHO_FHE_MSM_93.2.pdf;jsessionid=08EF4625562B4D043C61B201EEFE6C26?sequence=1

* keep in your mind: In newborns, the rectal temperature was equal to the axillary temperature plus 0.2 degrees C for each week of age up to 5 weeks. DOI: 10.1001/archpedi.1996.02170260078013

2- for Signs of Systemic Infection:

Our authors did not mention color change :

- Jaundice other than physiologic one (prolonged and deep color)

- Cyanosis (central)

- Skin rash

- excessive losing Body weight compared to birth weight, which is a sign of underlying pathology vs. malnutrition.

? please explain

6. PLOS authors have the option to publish the peer review history of their article (what does this mean?). If published, this will include your full peer review and any attached files.

Reviewer #1: No

Reviewer #2: No

---

## [Author Response · Author response to Decision Letter 0]

20 Jan 2021

We have responded to point by point reviewers' comments in the attached rebuttal letter.

---

## [Decision Letter · Decision Letter 1]

8 Feb 2021

Prevalence of clinical signs of possible serious bacterial infection and mortality associated with them from population-based surveillance of young infants from birth to 2 months of age

PONE-D-20-16849R1

Dear Dr. Yasir,

We’re pleased to inform you that your manuscript has been judged scientifically suitable for publication and will be formally accepted for publication once it meets all outstanding technical requirements.

Kind regards,

Claudia Marotta

Academic Editor

PLOS ONE

Additional Editor Comments (optional):

dear Authors congratulations

Reviewers' comments:

Reviewer's Responses to Questions

**Comments to the Author**

1. If the authors have adequately addressed your comments raised in a previous round of review and you feel that this manuscript is now acceptable for publication, you may indicate that here to bypass the “Comments to the Author” section, enter your conflict of interest statement in the “Confidential to Editor” section, and submit your "Accept" recommendation.

Reviewer #1: All comments have been addressed

Reviewer #2: All comments have been addressed

2. Is the manuscript technically sound, and do the data support the conclusions?

Reviewer #1: Yes

Reviewer #2: (No Response)

3. Has the statistical analysis been performed appropriately and rigorously? 

Reviewer #1: Yes

Reviewer #2: (No Response)

4. Have the authors made all data underlying the findings in their manuscript fully available?

Reviewer #1: Yes

Reviewer #2: (No Response)

5. Is the manuscript presented in an intelligible fashion and written in standard English?

Reviewer #1: Yes

Reviewer #2: Yes

6. Review Comments to the Author

Reviewer #1: Authors improved they already good paper. I find a very good manuscript and agree to publish it

The research topic is relevant and also the setting of study

Reviewer #2: (No Response)

7. PLOS authors have the option to publish the peer review history of their article (what does this mean?). If published, this will include your full peer review and any attached files.

Reviewer #1: No

Reviewer #2: No

---

## [Editor Report · Acceptance letter]

9 Feb 2021

PONE-D-20-16849R1 

Prevalence of clinical signs of possible serious bacterial infection and mortality associated with them from population-based surveillance of young infants from birth to 2 months of age 

Dear Dr. Nisar:

I'm pleased to inform you that your manuscript has been deemed suitable for publication in PLOS ONE. Congratulations! Your manuscript is now with our production department. 

Kind regards, 

on behalf of

Dr. Claudia Marotta 

Academic Editor

PLOS ONE